

# Mechanism of endothelial nitric oxide synthase phosphorylation and activation by tentacle extract from the jellyfish *Cyanea capillata*

Beilei Wang[1,2,*], Dan Liu[2,*], Chao Wang[2,*], Qianqian Wang[1,2], Hui Zhang[2], Guoyan Liu[1,2], Xia Tao[3] and Liming Zhang[1,2]

[1] Marine Bio-pharmaceutical Institute, Second Military Medical University, Shanghai, China
[2] Department of Marine Biotechnology, Faculty of Naval Medicine, Second Military Medical University, Shanghai, China
[3] Department of Pharmacy, Changzheng Hospital, Second Military Medical University, Shanghai, China
[*] These authors contributed equally to this work.

Corresponding authors
Xia Tao, taoxia2003@126.com
Liming Zhang,
lmzhang@smmu.edu.cn

## ABSTRACT

Our previous study demonstrated that tentacle extract (TE) from the jellyfish *Cyanea capillata* (*C. capillata*) could cause a weak relaxation response mediated by nitric oxide (NO) using isolated aorta rings. However, the intracellular mechanisms of TE-induced vasodilation remain unclear. Thus, this study was conducted to examine the role of TE on Akt/eNOS/NO and $Ca^{2+}$ signaling pathways in human umbilical vein endothelial cells (HUVECs). Our results showed that TE induced dose- and time-dependent increases of eNOS activity and NO production. And TE also induced Akt and eNOS phosphorylation in HUVECs. However, treatment with specific PI3-kinase inhibitor (Wortmannin) significantly inhibited the increases in NO production and Akt/eNOS phosphorylation. In addition, TE also stimulated an increase in the intracellular $Ca^{2+}$ concentration ($[Ca^{2+}]_i$), which was significantly attenuated by either $IP_3$ receptor blocker (Heparin) or PKC inhibitor (PKC 412). In contrast, extracellular $Ca^{2+}$-free, L-type calcium channel blocker (Nifedipine), or PKA inhibitor (H89) had no influence on the $[Ca^{2+}]_i$ elevation. Since calcium ions also play a critical role in stimulating eNOS activity, we next explored the role of $Ca^{2+}$ in TE-induced Akt/eNOS activation. In consistent with the attenuation of $[Ca^{2+}]_i$ elevation, we found that Akt/eNOS phosphorylation was also dramatically decreased by Heparin or PKC 412, but not affected by Nifedipine or H89. However, the phosphorylation level could also be decreased by the removal of extracellular calcium. Taken together, our findings indicated that TE-induced eNOS phosphorylation and activation were mainly through PI3K/Akt-dependent, PKC/$IP_3$R-sensitive and $Ca^{2+}$-dependent pathways.

## INTRODUCTION

Hypertension is the leading risk factor for cardiovascular diseases, causing almost 3.7% of total disability-adjusted life-years and 13% of all deaths (*Park, Kario & Wang, 2015*), which

has also been a serious public health burden all over the world. Thus, there is a critical need for developing anti-hypertensive drugs to control the blood pressure. However, many synthetic antihypertensive drugs are confirmed to have certain side effects, such as dizziness, headache, coughing, angioedema, abnormal taste, and kidney and liver problems (*Kuhlen & Forcucci, 2012*). Therefore, it is necessary to develop safer, more economical and innovative alternatives for the prevention or treatment of hypertension.

Currently, many bioactive natural products, especially marine ones, have received considerable attentions. Compared with terrestrial counterparts, marine organisms evolved a stronger biological activity and a more complex structural diversity to adapt to the extreme marine environment, such as high salt, high pressure, low-nutrient and unstable temperature condition (*Suleria et al., 2015*). Therefore, marine natural products may become a novel pharmaceutical resource to prevent and treat various diseases. In recent years, it was reported that protein hydrolysates derived from several marine organisms, such as cod (*Kim et al., 2000*), salmon (*Ono et al., 2006*), sea cucumber collagen (*Zhao et al., 2009*), shrimp (*Zhang et al., 2009*), sesame (*Biswas, Dhar & Ghosh, 2010*), squid skin (*Lin, Shun & Li, 2011*), and jellyfish (*Li et al., 2014*; *Liu et al., 2012*; *Zhuang et al., 2012*), could exert their hypotensive effects. Among them, jellyfish is one of the most abundant resources in marine ecosystems and may provide many promising sources of marine pharmaceuticals. In fact, the medicinal value of jellyfish has also been explored by Chinese people for a long time. It is believed to be effective to patients with hypertension, arthritis, ulcers and back pain. Besides, jellyfish can stimulate blood flow and reduce various types of swellings (*Liu et al., 2013*; *Liu et al., 2012*). Furthermore, some jellyfish-derived proteins have also been reported to possess antihypertensive (*Li et al., 2014*; *Liu et al., 2012*; *Zhuang et al., 2012*), antioxidant (*Bruschetta et al., 2014*), antimicrobial (*Ayed et al., 2012*) and anticoagulant activities (*Liu et al., 2015*; *Noguchi et al., 2005*). Although jellyfish is traditionally recognized to be beneficial in reducing blood pressure in China, its antihypertensive effect is rarely reported.

In our previous study, we demonstrated that tentacle extract (TE) from the jellyfish *Cyanea capillata* (*C. capillata*) could cause a weak relaxation response in isolated aorta rings, which might be mediated by nitric oxide (NO) (*Wang et al., 2013a*). However, the intracellular mechanism of TE-induced vasodilation and its molecular cross-talk remain unclear. It is well known that NO is produced by endothelial nitric oxide synthase (eNOS) in vascular endothelial cells (*Srivastava, Bath & Bayraktutan, 2012*). eNOS is a calcium dependent enzyme and activated by the increase in intracellular free calcium concentration ($[Ca^{2+}]_i$) (*Chen et al., 2010*; *Kerr et al., 2012*), which is induced either by an influx of extracellular calcium via channels such as voltage-dependent calcium channels, or by the release from intracellular stores in endoplasmic reticulum (ER) via receptors such as inositol 1,4,5-triphosphate receptors ($IP_3Rs$) (*Sammels et al., 2010*). Besides, the activity of eNOS is also regulated by phosphorylation level. For example, eNOS phosphorylation at Ser1177 by phosphatidylinositol 3-kinase (PI3-K)-dependent Akt plays a critical role in eNOS activation (*Yoshitomi et al., 2011*). Thus, the current study was designed to investigate the effects and molecular mechanisms of TE on eNOS activity in endothelial

cells by detecting the changes in both intracellular $Ca^{2+}$ concentration and Akt-dependent signal transduction pathways.

## MATERIALS AND METHODS

### Drugs and chemicals reagents

The HUVECs cell line was purchased from Zhongqiaoxinzhou Biotech (Shanghai, China). MTT assay kit and NOS assay kit were purchased from Beyotime (Jiangsu, China). Human NO ELISA assay kit was purchased from Sangon Biotech (Shanghai, China). eNOS inhibitor $N^{\omega}$-nitro-L-arginine methyl ester (L-NAME) and PI3-K inhibitor Wortmannin were purchased from Sigma-Aldrich (St. Louis, MO, USA). The antibodies against phospho-Akt (Ser473), Akt, phospho-eNOS (Ser1177) and eNOS were purchased from Cell Signaling Technology (Beverly, MA, USA). The antibody against GAPDH was purchased from Abcam (Cambridge, MA, USA). HRP-conjugated anti-rabbit IgG and anti-mouse IgG were purchased from Beyotime (Jiangsu, China). Fluo-4 AM was purchased from Invitrogen (Carlsbad, CA, USA). The stock solution of 1 mM was prepared by adding dimethyl sulfoxide (DMSO) to solid powder. The working solution of 5 µM was prepared by adding serum free medium to the stock solution. $1 \times$ HBSS (without phenol red, liquid, sterile-filtered) and $1 \times$ HBSS (without phenol red,1.26 mM $CaCl_2$) were purchased from Sangon Biotech (Shanghai, China). L-type calcium channel blocker Nifedipine, PKA inhibitor H89, $IP_3$ receptor blocker Heparin and PKC inhibitor PKC 412 were purchased from Sigma-Aldrich (St. Louis, MO, USA). In the measurement of $Ca^{2+}$ mobilization, Nifedipine solution of 100 µM was prepared by $Ca^{2+}$-containing HBSS, whereas H89 solution of 10 µM, Heparin solution of 125 IU and PKC 412 solution of 10 µM were prepared by $Ca^{2+}$-free HBSS.

### TE preparation from the jellyfish *C. capillata*

Specimens of *C. capillata* were collected in June, 2014, in the Sanmen Bay, East China Sea, and identified by Professor Huixin Hong from the Fisheries College of Jimei University, Xiamen, China. The removed tentacles were preserved in plastic bags on dry ice and immediately shipped to Shanghai, where the samples were frozen at $-70\,°C$ until use. TE was prepared following the method as described in previous reports (*Bloom, Burnett & Alderslade, 1998*; *Wang et al., 2013b*). Briefly, frozen tentacles were thawed at $4\,°C$ and immersed in filtered seawater at a mass/volume ratio of 1:1 to allow autolysis of the tissues for four days. The mixture was stirred for 30 min twice daily. The autolyzed mixture was centrifuged at $10,000 \times g$ for 15 min, thrice. The resultant supernatant was the TE. All procedures were performed at $4\,°C$ or in an ice bath. The TE was centrifuged at $10,000 \times g$ for 15 min to remove the sediments, followed by dialysis against phosphate buffered saline (PBS, 0.01 mol/L, pH 7.4) for over 8 h before use. The protein concentration in the preparations was determined using the method of Bradford.

### Endothelial cell cultures

Human umbilical vein endothelial cells (HUVECs) were cultured in high glucose DMEM (Hyclone, Waltham, MA, USA) supplemented with 10% fetal bovine serum (FBS, Gibco,

Carlsbad, CA, USA), 100 U/ml penicillin and 100 μg/ml streptomycin at 37 °C in a humidified incubator with 95% air and 5% $CO_2$.

## Cell viability assay

Cell viability was determined by the MTT assay. Cells were plated in 96-well culture plates at a density of $10^4$ cells/ml. After incubation for 24 h, cell groups were respectively treated with various doses of TE (0–24 μg/ml) for 1 h or 6 h. Then, 20 μl MTT reagent (5 mg/ml) was added to each well. At 4 h later, the supernatants were removed and the formazan dye was subsequently dissolved in DMSO. The absorbance value at 490 nm was measured using a microplate reader (BioTek, Winooski, VT, USA).

## Measurement of eNOS activity in HUVECs

Cells were serum-starved overnight in 96-well culture plates before measurements. In the dose–effect experiments, cells were incubated with different concentrations of TE (0–4 μg/ml) at 37 °C for 1 h. In the time-effect experiments, cells were incubated with TE (1 μg/ml) for different time durations (0–180 min) at 37 °C. After treatments, eNOS activity was measured according to manufacturer's instructions (NOS assay kit). Briefly, cells were exposed to 100 μl reaction buffer solutions and subsequent 100 μl reaction solutions for 2 h with/without eNOS inhibitor (L-NAME). Then the plates were observed on a microplate reader (BioTek, Winooski, VT, USA) at excitation/emission wavelengths of 495/515 nm. The activity of eNOS was calculated by the following equation: relative activity (eNOS) = $(\text{RFU}_{stimulated} - \text{RFU}_{inhibitor+stimulated})/(\text{RFU}_{unstimulated} - \text{RFU}_{inhibitor+unstimulated})$.

## Measurement of NO concentration in HUVECs

HUVECs were divided into three groups. In the dose–effect experiments, cells were incubated with different concentrations of TE (0–4 μg/ml) at 37 °C for 1 h. In the time-effect experiments, cells were incubated with TE (1 μg/ml) for different time durations (0–180 min) at 37 °C. In the third group, cells were incubated with eNOS inhibitor L-NAME or PI3-K inhibitor Wortmannin for 15 min before treatment with TE (1 μg/ml, 1 h). After incubation, culture supernatants were collected, and NO concentration was measured using a microplate reader (BioTek, Winooski, VT, USA) in the absorbance value at 450 nm according to manufacturer's instructions (Human NO ELISA assay kit).

## Western blotting

In the time-effect group, HUVECs were treated with TE (1 μg/ml) for different time durations (0-60 min). In PI3K/Akt-dependent signaling pathways, HUVECs were treated with TE (1 μg/ml) for 30 min in the presence or absence of Wortmannin. In calcium-dependent signaling pathways, HUVECs were treated with TE (1 μg/ml) for 30 min in the presence of extracellular $Ca^{2+}$-containing, Nifedipine, extracellular $Ca^{2+}$-free, H89, Heparin and PKC 412, respectively.

After treatments, cells were lysed on ice in RIPA buffer with protease inhibitor (1% PMSF). The protein content of the lysate was measured by the method of Bradford. Equal amounts of protein per sample were loaded on 10% SDS-PAGE gels and then transferred to the nitrocellulose membranes. The membranes were subsequently blocked with 5% non-fat

dry milk in TBST (3 g Tris-base, 8 g NaCl, 0.2 g KCl, 0.05% Tween-20, dilute with water to 1,000 ml, pH 7.4) for 2 h at room temperature. Then, the samples were incubated overnight at 4 °C with primary antibodies as follows: p-Akt (1:1,000), Akt (1:1,000), p-eNOS (1:500), eNOS (1:500) and GAPDH (1: 5,000), with gentle shaking. The ECL method was used with secondary antibodies (HRP-conjugated anti-rabbit IgG and anti-mouse IgG) at a dilution of 1:5,000 for 2 h at room temperature. After that, membranes were exposed using a chemiluminescent detection system (Syngene G: Box; Syngene, Cambridge, UK). Quantitative densitometric analyses of immunoblots were performed using an ImageJ software (Ver. 1.48), and the relative ratio was calculated.

## Measurement of $Ca^{2+}$ mobilization

HUVECs were cultured in a confocal dish (Coverglass Bottom Dish; Corning, Inc., Corning, NY, USA) and serum-starved overnight before use. Cells were loaded with Fluo-4 AM working solution (5 μM) at 37 °C in the dark for 30 min, then washed three times with $Ca^{2+}$-free HBSS to remove excess extracellular dye.

To characterize the possible contribution of extracellular calcium, the cells were pre-treated with 1.26 mM $Ca^{2+}$-containing HBSS with/without Nifedipine (100 μM) for 15 min before the addition of TE (10 μg/ml final concentration). To examine the possible participation of intracellular calcium stores, the cells were pre-treated with $Ca^{2+}$-free HBSS in the presence or absence of H89 (10 μM), Heparin (125 IU) or PKC 412 (10 μM) for 15 min before the addition of TE (10 μg/ml final concentration). All the concentrations of the inhibitors were screened out and proved effective in similar preparations of the preliminary experiments.

The fluorescence of $Ca^{2+}$ mobilization in HUVECs were monitored using a laser scanning confocal microscope (Olympus FV 1000; Olympus, Tokyo, Japan) with excitation and emission wavelengths of 488 nm and 526 nm, respectively. Fluorescence images were recorded as time-series mode per 10 s intervals (100 images altogether). TE was added after the first five images were collected. Fluorescence intensities were obtained from the data set of images using FV10-ASW 3.1 (Olympus, Tokyo, Japan).

## Statistical analysis

In the experiments, the images shown were the representatives of at least three experiments performed on different experimental days. Data were presented as mean ± SEM. Analysis of variance (ANOVA) and Student's $t$-test were used in statistical evaluation of the data as appropriate. $P$-values less than or equal to 0.05 were considered significant.

## RESULTS

### Effects of TE on the viability of HUVECs

As shown in Fig. 1, when HUVECs were treated with various doses of TE for 1 h, the viability was decreased at the concentrations of 12–24 μg/ml in a dose-dependent manner, and the $IC_{50}$ was 15.55 μg/ml. However, TE at 0.5–8 μg/ml, did not significantly influence the cell viability after 1 h treatment. Similarly, after treatment with TE for 6 h, the viability of cells was decreased at the concentrations of 4–24 μg/ml in a dose-dependent manner,

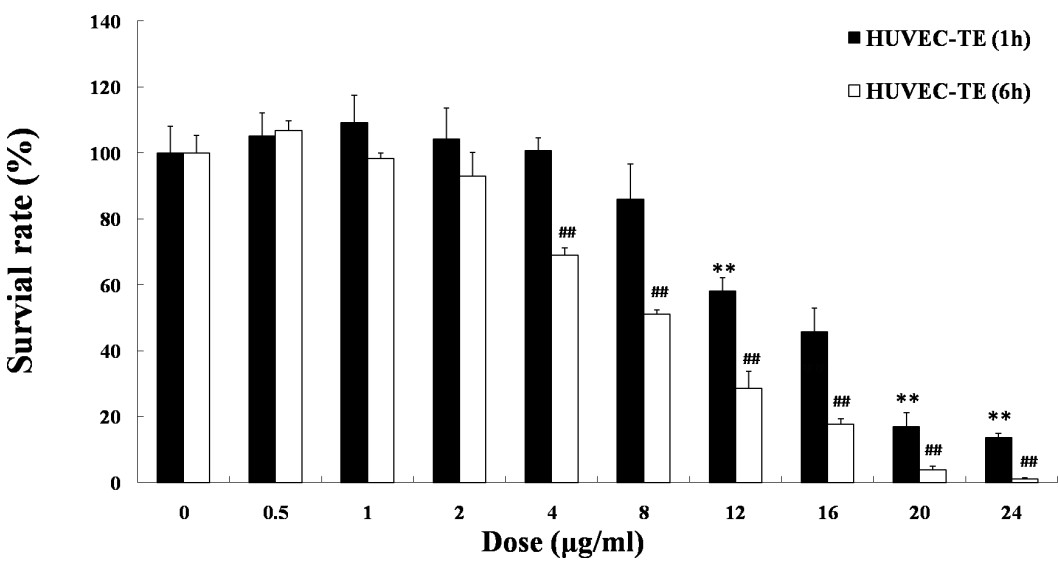

**Figure 1** **Effects of TE on the viability of HUVECs. HUVECs were treated with the following agents: 0–24 μg/ml of TE.** The MTT assays were performed at 1 h and 6 h after treatments. Each column represents mean ± SD of six samples. **$P < 0.01$ *vs.* Control (1 h), ##$P < 0.01$ *vs.* Control (6 h).

and the $IC_{50}$ was 14.88 μg/ml. But TE at 0.5–2 μg/ml did not significantly influence the cell viability, either. Therefore, the relatively safe dose of TE less than 8 μg/ml was selected for the next study.

## Effect of TE on the eNOS activity and NO production in HUVECs

Treatments with various doses of TE (0–4 μg/ml) increased the activity of eNOS in a concentration-dependent manner within 1 h in HUVECs: a significant increase was induced at the concentration of 0.5 μg/ml and the peak at 4 μg/ml (Fig. 2A). Similarly, TE-induced NO production was also concentration-dependent within 1 h: a significant increase was induced at the concentration of 1 μg/ml and the peak at 4 μg/ml (Fig. 2C). Therefore, the dose of 1 μg/ml was used in the next time-effect experiments. It was also showed a significant time-dependent increase was induced by TE (1 μg/ml) in the activity of eNOS (Fig. 2B) and NO production (Fig. 2D), which reached the maximum at about 1 h and maintained until 3 h. Therefore, the concentration of 1 μg/ml and the time of 1 h were selected for the next study on PI3K/Akt/eNOS signaling pathways.

## Effect of TE on the PI3K/Akt/eNOS signaling pathway in HUVECs

We next examined the phosphorylation of Akt and eNOS after the stimulation with TE (1 μg/ml). As shown in Fig. 3A, Akt phosphorylation was significantly induced by TE from 15 to 45 min, with the maximum phosphorylation occurring at about 15 min, while TE seemed to have no effect on the level of total Akt. Similarly, eNOS phosphorylation was also induced by TE from 15 to 60 min, with the maximum phosphorylation at about 30 min, while TE had no effect on the level of total eNOS, either. These results indicated that TE could activate the Akt/eNOS pathways. On the other hand, since Akt had been reported to phosphorylate eNOS via PI3-kinase, we then pretreated HUVECs with the PI3-K inhibitor
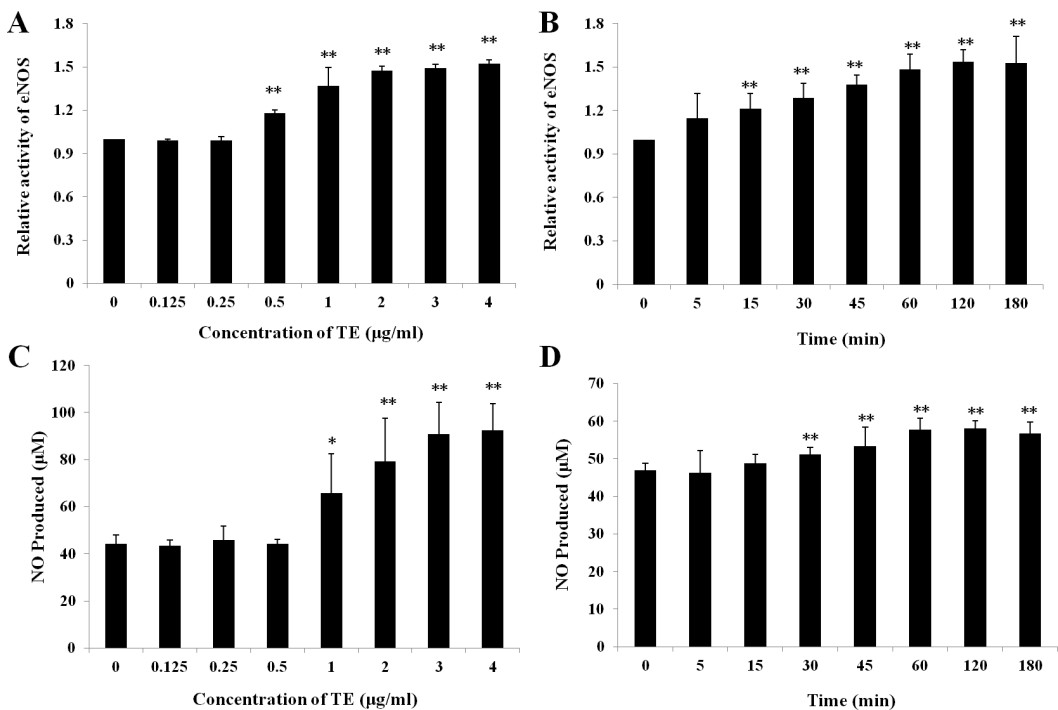

**Figure 2 Effects of TE on relative activity of eNOS and NO production in HUVECs.** (A) and (C) HU-VECs were treated with various dose of TE (0–4 μg/ml) for 1 h. (B) and (D) HUVECs were treated with TE (1 μg/ml) for various time (0–180 min). Each column represents mean ± SD of six samples. $^{**}P <$ 0.01, $^{*}P < 0.05$ *vs.* Control.

Wortmannin (10 μM) before TE to investigate whether this upstream signaling pathway was involved. As shown in Fig. 3B, the inhibition of PI3-kinase completely blocked the TE-induced Akt/eNOS phosphorylation, thus demonstrating the requirement for this kinase during the Akt/eNOS activation by TE.

## Effect of PI3K/Akt/eNOS pathway inhibition on the TE-induced NO production

As shown in Fig. 4, both eNOS inhibitor L-NAME and PI3-K inhibitor Wortmannin resulted in a significant reduction in TE-induced NO production, which indicated that PI3K/Akt/eNOS mediated the release of NO induced by TE, suggesting an important role of the PI3K/Akt/eNOS pathway in TE-induced NO release.

## Effects of TE on intracellular Ca$^{2+}$ concentration changes in HUVECs

To explore the effects of TE upon the Ca$^{2+}$ concentration in HUVECs, we incubated the cells with the Ca$^{2+}$ indicator dye Fluo-4 AM, and then stimulated them with TE whilst the time-series images of intracellular Ca$^{2+}$ levels were detected. As shown in Fig. 5, TE could induce a rapid rise in the intracellular Ca$^{2+}$ concentration ([Ca$^{2+}$]$_i$), which peaked within 1 min and sustained for about 10 min. Secondly, to determine the involvement of extracellular calcium, we next used the L-type calcium blocker Nifedipine (in Ca$^{2+}$-containing HBSS) and Ca$^{2+}$-free HBSS instead of the normal HBSS for the examination of TE-induced [Ca$^{2+}$]$_i$ changes. Our results showed that neither Nifedipine nor extracellular Ca$^{2+}$-free

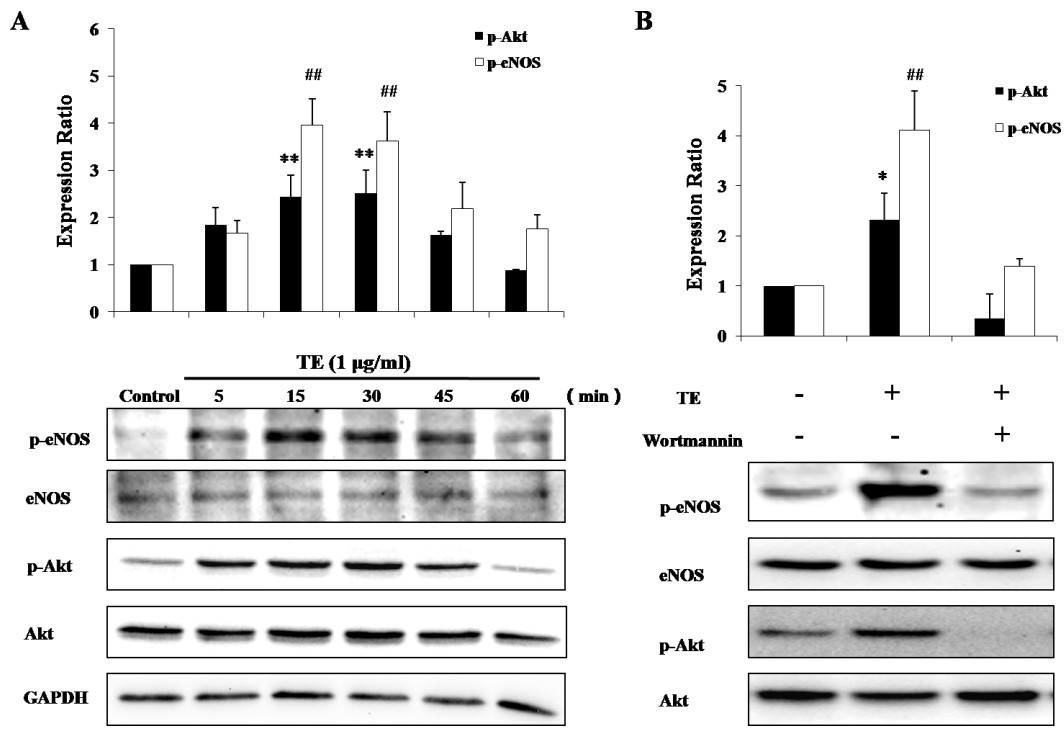

**Figure 3** **Responses of the PI3K/Akt/eNOS signaling pathway to TE (1 µg/ml).** (A) HUVECs were treated with TE (1 µg/ml) for various time durations. (B) HUVECs were treated with TE (1 µg/ml) in the presence or absence of Wortmannin (PI3-K inhibitor) for 30 min. $^*P < 0.05$, $^{**}P < 0.01$ vs. p-Akt/Akt of Control. $^{\#\#}P < 0.01$ vs. p-eNOS/eNOS of Control.

had influences on the TE-induced $[Ca^{2+}]_i$ rise, which indicated that TE-induced $[Ca^{2+}]_i$ rise did not come from the extracellular $Ca^{2+}$ influx, but mainly from the intracellular stored $Ca^{2+}$ release. Therefore, to further explore the source of $Ca^{2+}$, we next investigated the effects of PKA inhibitor H89, $IP_3$ receptor blocker Heparin and PKC inhibitor PKC 412 on the $[Ca^{2+}]_i$ elevation. Our results showed that the TE-induced $[Ca^{2+}]_i$ rise was significantly attenuated either by Heparin or by PKC 412, but not affected by H89, suggesting that the $IP_3R$ and PKC signaling play major roles in the TE-induced $[Ca^{2+}]_i$ elevation.

## Effect of calcium signaling on the TE-induced Akt/eNOS activation

Since eNOS is also activated by a rapid increase in the intracellular $Ca^{2+}$, we next explored the role of $Ca^{2+}$ in the activation of Akt/eNOS pathway in response to TE. As shown in Fig. 6, we indeed found the evidence for a critical role of $Ca^{2+}$ in the TE-induced Akt/eNOS activation. Firstly, we found that TE (in $Ca^{2+}$-containing HBSS) could induce the phosphorylation of both Akt and eNOS, while the phosphorylation level was dramatically decreased by the removal of extracellular calcium, indicating that $Ca^{2+}$ was essential to the TE-induced Akt/eNOS activation. Secondly, to further elucidate the contribution of extracellular and intracellular calcium to the effects of TE-induced Akt/eNOS activation, we similarly performed the Akt/eNOS activity assays in the presence of L-type calcium channel blocker Nifedipine (100 µM), PKA inhibitor H89 (10 µM), $IP_3$ receptor blockers Heparin (125 IU) and PKC inhibitor PKC 412 (10 µM), respectively. As shown in Fig. 6, the

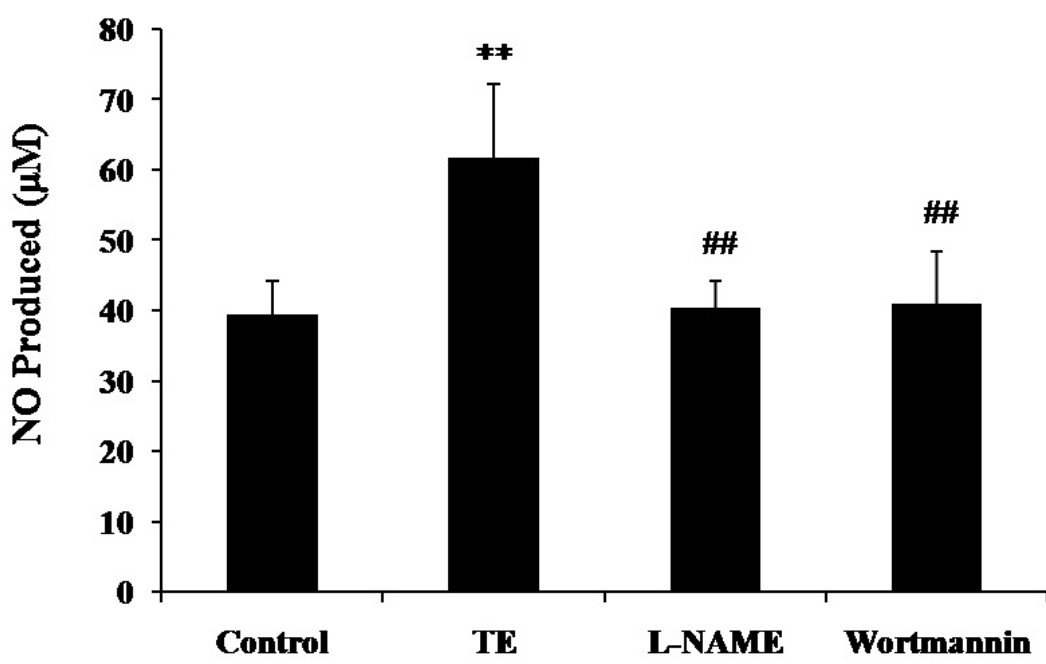

**Figure 4** **Effects of PI3K/Akt/eNOS inhibitors on TE-induced NO production.** HUVECs were treated with TE (1 μg/ml) in the presence or absence of eNOS inhibitor (L-NAME) or PI3-K inhibitor (Wortmannin) for 1 h. **$P < 0.01$ *vs.* Control, ##$P < 0.01$ *vs.* TE.

phosphorylation of Akt/eNOS was completely attenuated by the incubation with Heparin or PKC 412, which indicated that the Akt/eNOS activation might be associated with $Ca^{2+}$ release from the ER through $IP_3R$ and PKC pathways. This result was in consistent with that of the calcium fluorescence assay. However, the expressions of p-Akt/p-eNOS were not affected by Nifedipine or H89, which indicated that the Akt/eNOS activation was not due to the influx of extracellular $Ca^{2+}$ via L-type calcium channel or through PKA-dependent $Ca^{2+}$ signaling.

## DISCUSSION

In recent years, many research works have been conducted to explore pharmacological and cardiovascular characterization of jellyfish venoms. However, the current studies have been hindered by some reasons: (1) jellyfish venoms are sticky, thermolabile and difficult to be separated (*Feng et al., 2010*); and (2) venom samples are difficult to be collected due to the small amount of the venoms in nematocysts (*Xiao et al., 2009*). To solve the problem, we have previously compared the nematocyst venoms with TE (devoid of nematocysts) from the jellyfish *C. capillata*. Our result suggested that TE may serve as a potential alternative of the nematocyst venoms with much richer source for isolating and purifying cardiovascular active proteins (*Xiao et al., 2009*), because these proteins in the nematocyst venoms and TE are probably encoded by the same gene fragment (*Nagai et al., 2000a*; *Nagai et al., 2000b*). Besides, using isolated rat aortic rings, we also verified that TE did have a direct vascular activity. Our results showed that TE could cause a weak relaxation response, which was significantly attenuated either by the removal of the endothelium,
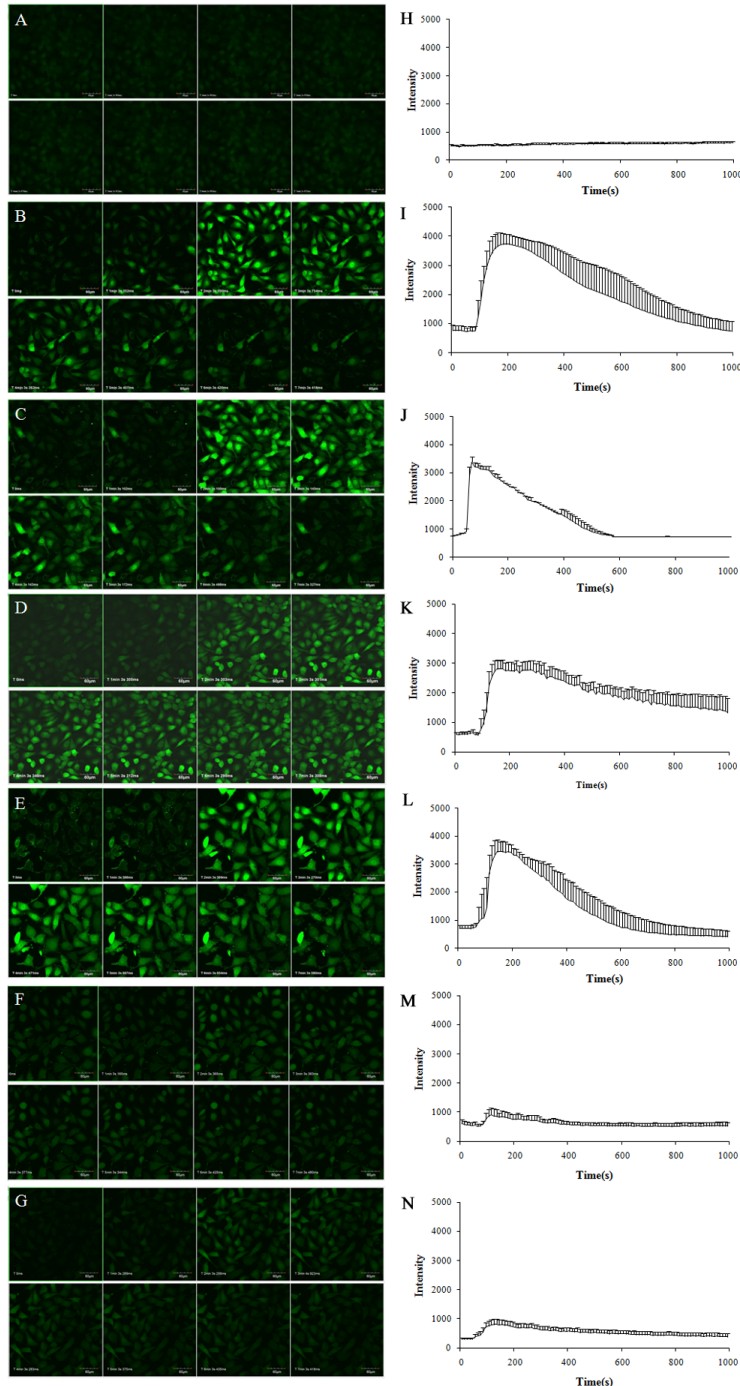

**Figure 5** **Characterization of the Ca$^{2+}$ concentration evoked by TE (10 µg/ml) in Fluo-4-loaded HUVECs.** (A–G) Experimental records of the Ca$^{2+}$ fluorescence image. (H–N) Statistical results of the F$_{488}$/F$_{526}$ ratio. (A, H) Control; (B, I) incubated with Ca$^{2+}$-containing HBSS; (C, J) incubated with Ca$^{2+}$-containing HBSS *plus* Nifedipine; (D, K) incubated with Ca$^{2+}$-free HBSS; (E, L) incubated with Ca$^{2+}$-free HBSS *plus* H89; (F, M) incubated with Ca$^{2+}$-free HBSS *plus* Heparin; (G, N) incubated with Ca$^{2+}$-free HBSS *plus* PKC 412.

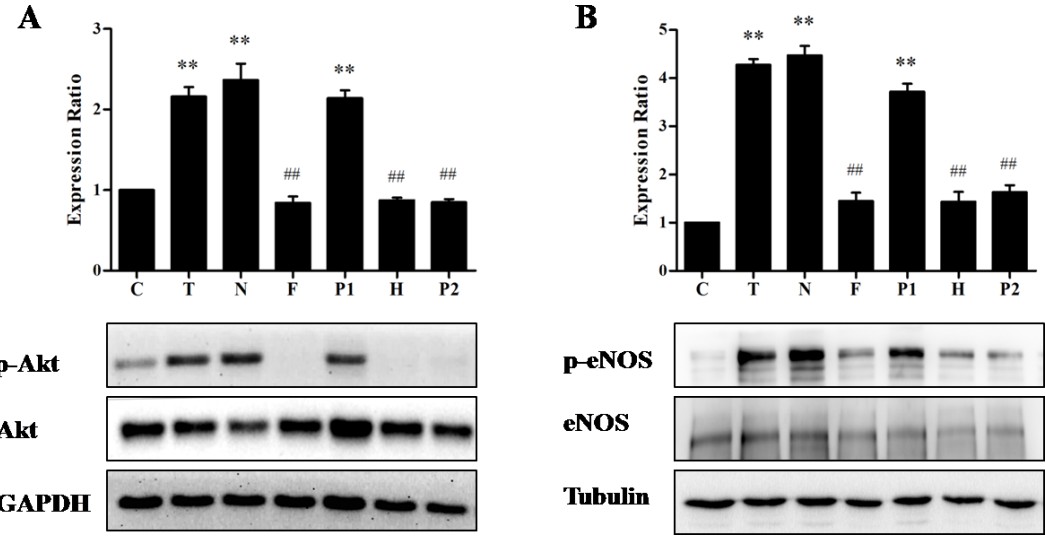

**Figure 6 Effects of Ca²⁺ signaling on TE-induced Akt/eNOS activation.** HUVECs were pretreated for 15 min with HBSS (Ca²⁺-containing), Nifedipine (in Ca²⁺-containing HBSS), HBSS (in Ca²⁺-free HBSS), H89 (in Ca²⁺-free HBSS), Heparin (in Ca²⁺-free HBSS), PKC 412 (in Ca²⁺-free HBSS), respectively, then stimulated with TE (1 μg/ml) for 30 min. (A) Akt and phospho-Akt (Ser473) were observed by immunoblotting with a phospho-specific antibody. (B) eNOS and phospho-eNOS (Ser1177) were assayed by immunoblotting with a phospho-specific antibody. **$P < 0.01$ *vs.* Control. ##$P < 0.01$ *vs.* TE (Ca²⁺). C: Control, T: TE (Ca²⁺), N: Nifedipine, F: TE (Ca²⁺-free), P1: PKA inhibitor (H89), H: Heparin, P2: PKC inhibitor (PKC 412).

or by the blockade of NO synthase by L-NAME (*Wang et al., 2013a*), suggesting that the vasodilation induced by TE was possibly mediated by an NO-dependent pathway. So in the current study, we subsequently measured the eNOS activity and NO concentration induced by TE in HUVECs. Our results showed that TE could induce concentration- and time-dependent increases in eNOS activity and NO production in HUVECs; in addition, eNOS inhibitor L-NAME completely attenuated NO production induced by TE, confirming that TE-induced vasodilative effects were mainly mediated by the release of NO via the activation of eNOS in the endothelial cells.

It is well-established that eNOS is tightly regulated not only at the transcriptional level but also by certain post-transcriptional mechanisms (*Vilahur et al., 2014*; *Yoshitomi et al., 2011*). In the present study, we found that TE did not have an effect on the level of eNOS protein, but did markedly induce eNOS phosphorylation at Ser1177 from 15 to 60 min, suggesting that TE induced an increase in eNOS activity at the post-transcriptional level in HUVECs. To further explore the possible mechanisms underlying eNOS activation in HUVECs after treatment with TE, we next investigated the potential role of PI3K/Akt-dependent signaling. Our results demonstrated that the PI3K/Akt pathway is necessary for eNOS activation in TE-treated HUVECs. Several lines of evidence supported this notion: (1) TE stimulated Akt phosphorylation from 15 to 45 min, which occurred slightly preceding eNOS phosphorylation; (2) PI3-kinase inhibitor Wortmannin, not only blocked TE-evoked Akt, but also inhibited TE-induced Ser1177 phosphorylation of eNOS under

the same condition; (3) Wortmannin completely attenuated TE-induced NO production, consistent with the effects of the eNOS inhibitor L-NAME.

On the other hand, since calcium ions play a crucial role in stimulating eNOS activity through a $Ca^{2+}$/calmodulin-dependent mechanism (*Chen et al., 2010*), we next explored the role of $Ca^{2+}$ in the activation of eNOS in response to TE. Firstly, we assessed the effects of TE on intracellular $[Ca^{2+}]_i$ in HUVECs using the calcium specific fluorescent dye, Fluo-4 AM, and found that TE did induce a rapid rise in the intracellular $[Ca^{2+}]_i$, which peaked within 1 min and sustained for about 10 min. Secondly, since $Ca^{2+}$ is maintained by two mechanisms: entry from extracellular medium through the opening of calcium permeable channels in plasma membrane, or release from intracellular organelles (mainly ER), we further explored the source of TE-evoked $Ca^{2+}$ rise using $Ca^{2+}$ signaling related inhibitors. Our results showed that neither Nifedipine nor extracellular $Ca^{2+}$-free had a significant influence on the TE-induced $[Ca^{2+}]_i$ rise, indicating that TE-induced $[Ca^{2+}]_i$ rise did not come from the extracellular $Ca^{2+}$ influx, but mainly from the intracellular stored $Ca^{2+}$ release. In ER, the $IP_3R$ channel is capable of releasing a large quantity of $Ca^{2+}$ to the cytosol, and is believed to play a primary role in $Ca^{2+}$ mobilization (*Morgado et al., 2012*; *Tiruppathi et al., 2002*). This study also showed that $Ca^{2+}$ elevation was significantly attenuated by the blockade of $IP_3R$ by Heparin, suggesting that intracellular stored $Ca^{2+}$ release via $IP_3R$ played major roles in the TE-induced $[Ca^{2+}]_i$ elevation. On the other hand, it was also reported that the $IP_3R$ could be phosphorylated by various protein kinases, such as PKA and PKC, thus its function might be modulated by these kinases (*Morgado et al., 2012*). So we next investigated the dependence of PKA and PKC on the $[Ca^{2+}]_i$ elevation. Our results showed that the TE-induced $[Ca^{2+}]_i$ rise was significantly attenuated by PKC 412, but not affected by H89, suggesting that the PKC pathway might act on $IP_3R$ and then cause $Ca^{2+}$ release, while the PKA pathway seemed to be ineffective in stimulating the $IP_3R$. After that, we further tested the role of $Ca^{2+}$ in the activation of Akt/eNOS pathway in response to TE. Firstly, our results showed that TE (in $Ca^{2+}$-containing HBSS) did induce the phosphorylation of both Akt and eNOS, while the phosphorylation level was dramatically decreased by the removal of extracellular calcium, indicating that extracellular $Ca^{2+}$ was essential to TE-induced eNOS activation. Since we had confirmed that TE-induced $Ca^{2+}$ mainly came from intracellular stored $Ca^{2+}$ rather than extracellular $Ca^{2+}$, we hypothesized that extracellular calcium might be necessary for Akt phosphorylation, and subsequently activate eNOS. In fact, it has been reported that calcium could phosphorylate three kinases (Akt, Erk and Fak) that are involved in the cell survival signalling in neuroblastoma (*Satheesh & Busselberg, 2015*). However, to determine whether these effects are direct or not, and to clarify the exact effects of extracellular $Ca^{2+}$ on Akt, further investigations are required. Secondly, we found that TE-induced Akt/eNOS activation was not affected by Nifedipine, suggesting that $Ca^{2+}$ influx via L-type calcium channels was not involved in TE-evoked Akt/eNOS activation, which was consistent with the results of calcium fluorescent assay where Nifedipine had no influence on the TE-induced $[Ca^{2+}]_i$ rise. Finally, the phosphorylation of Akt/eNOS was significantly attenuated by the inhibition of $IP_3R$ (Heparin) or PKC (PKC 412), but not affected by PKA inhibitor (H89), which was also in line with the results in fluorescence assay, indicating that TE-induced Akt/eNOS

activation might be associated with the calcium release via IP$_3$R, while IP$_3$R was also modulated by PKC pathways rather than PKA pathways. Taken together, these findings confirmed that calcium pathway was also necessary in the activation of TE-induced Akt/eNOS signaling.

## CONCLUSIONS

In this study, we demonstrated that TE from *C. capillata* could induce dose- and time-dependent activation of eNOS and NO production. Further investigation found that TE induced Ser1177 eNOS phosphorylation and activation mainly through PI3K/Akt-dependent, PKC/IP$_3$R-sensitive and Ca$^{2+}$-dependent pathway. Since dysfunction of endothelial NO production is one of the major predictors of cardiovascular events, these findings will contribute to a better understanding of the signaling mechanisms of TE in regulating the endothelial function. Although TE may require further purification and identification in the near future, the current study opens up the possibilities for the development of jellyfish-derived specific-acting drugs that can be used to treat and/or prevent cardiovascular diseases such as hypertension.

## ACKNOWLEDGEMENTS

The authors thank Pro. Huixin Hong from the Fisheries College of Jimei University for his careful identification of the jellyfish species and Mr. Fang Wei from the Foreign Languages' Office of the Second Military Medical University for his careful revision of the English language of the manuscript.

### Funding

This work was supported by the Young Scientists Fund of the National Natural Science Foundation of China (41506178), the Young Scientists Fund of the National Natural Science Foundation of China (81401578), the National Natural Science Foundation of China (81370833) and the National Major Scientific and Technological Special Project for "Significant New Drugs Development" (Ministry of Science and Technology) (2013ZX09J13110-07B). The funders had no role in study design, data collection and analysis, decision to publish, or preparation of the manuscript.

### Grant Disclosures

The following grant information was disclosed by the authors:
Young Scientists Fund of the National Natural Science Foundation of China: 41506178, 81401578.
National Natural Science Foundation of China: 81370833.
Ministry of Science and Technology: 2013ZX09J13110-07B.

### Competing Interests

The authors declare there are no competing interests.

## Author Contributions

- Beilei Wang conceived and designed the experiments, performed the experiments, wrote the paper.
- Dan Liu and Chao Wang performed the experiments, analyzed the data, wrote the paper, prepared figures and/or tables.
- Qianqian Wang, Hui Zhang and Guoyan Liu contributed reagents/materials/analysis tools.
- Xia Tao reviewed drafts of the paper.
- Liming Zhang conceived and designed the experiments, reviewed drafts of the paper.

## Data Availability

The raw data has been supplied as a Supplementary File.

## Supplemental Information

Supplemental information for this article can be found online at http://dx.doi.org/10.7717/peerj.3172#supplemental-information.

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
