# Peer review of "Mechanism of endothelial nitric oxide synthase phosphorylation and activation by tentacle extract from the jellyfish Cyanea capillata"

_PeerJ, doi:10.7717/peerj.3172_

## Round 0.1 · original submission · Major Revisions

· Academic Editor

Major Revisions

The editor's comments and reviews:
1. Previously the authors reported the TE has basic pharmacological activity.
In this study, they report the eNOS and related events-activating properties of the TE without fundamental informtstion on the ingredients or components if TE.
Because this is the second manuscript of the TE pharmacology, the structural and chemical compounds or molecules of the TE are necessary for justification of their findings. Therefore, the additional structural studies are required during revision,

Reviewer 1 ·

Basic reporting

This paper is very interesting and deals with some important aspects of pharmacological studies of the jellyfish tentacle extract (TE) on vascular endothelial cells. I think that this paper is suitable for “Peer J” and fits well in the aim and scope of the journal. In my opinion, the manuscript can be accepted for publication after minor revision and after the indicated explanations and corrections.

Experimental design

On the whole, the manuscript is well-written and the experiments are well-planned and executed.

Validity of the findings

A major concern is that the lack of discussion about the TE, which would have no nematocyst content (and thus no venom) due to the techniques used for TE preparation - the nematocysts would have been discarded in the pellet after centrifugation. However, in Fig.1, TE could induce a significant dose-dependent decrease of the cell viability by MTT assay. The Authors should explain this result and provide the definition of the term "viability".

Additional comments

1. Calcium concentrations under the Ca2+-containing conditions should be stated in Methods section.
2. I suggest the authors to make a step forward and try to isolate the single component (such as protein or peptide) responsible for the observed effects in this study from the tentacle extract of C. capilata. A purified component and its activity would give a stronger evidence of the observed effects in this study.

Reviewer 2 ·

Basic reporting

The paper is well written

Experimental design

Major concern:
Human umbilical vein endothelial cells (HUVECs) contributes greatly to vascular and blood pressure homeostasis. Nitric oxide is crucial for vasodilation, preventing leucocyte adhesion and extravasation. It is well known that primary cultures of endothelial cells require specific growth supplement (i.e. Journal of Immunological Methods 254 2001 183–190), thus, the authors need to explain why in their cell culture conditions any specific growth supplements were used. In this view, a morphologic study, confirming the maintenance of endothelial cells phenotype during different experimental procedures might be necessary.

Validity of the findings

Minor concern:
Figure 2, panel C and D, there is a discrepancy between NO production in HUVECs after 1h treatment for 1 hour. In panel C, NO production reached 4 microM while in panel D, in the same conditions, NO production reached 50 microM. Please explain

Additional comments

no comment

Reviewer 3 ·

Basic reporting

Overall, the English languge of this article is ok. But, it still needs some polishing and corrections of the typos.
The introduction and backgound of this article well demonstrates why the authors try to develop safer and more innovative alternatives from jellyfish tentacle extract of Cyanea capillata for the prevention or treatment of hypertension.
The structure of the article is acceptable. But, some figs need much more clear, such as Fig.5 right panel.

Experimental design

The experimental design of this study is proper to to examine the role of TE on Akt/eNOS/NO and 25 Ca2+ signaling pathways in human umbilical vein endothelial cells (HUVECs). But,can authors explain why they used the TE instead of the jellyfish nematocyst venom in this study?

Validity of the findings

The findings of the article is good and the data is robust with well statistical analysis. But, can author explain why the responses of PI3K/Akt/eNOS signaling pathway to TE vary after being treated with 1 μg/ml TE for various time durations and what is the reason for the decrease of expression ratio more than 15 min?

Additional comments

The mechanism of endothelial nitric oxide synthase phosphorylation and activation by tentacle extract from the jellyfish Cyanea capillata is reported in this paper. This work is very interesting and important. The results is also very clear to demonstrate that the TE-induced eNOS phosphorylation and activation are mainly through
39 PI3K/Akt-dependent, PKC/IP3R-sensitive and Ca2+-dependent pathways. Although, there are still some explainations and corrections needed. I think, this article can be accepted if the authors make some revisions.

---

## Round 0.2 · Major Revisions

· Academic Editor

Major Revisions

The revision has been performed carefully.

However, the present revision is still not satisfactory with regard to the basic material.

The jellyfish tentacle extract has well been documented with the venom properties and peptide molecules. The present authors examined the tentacle extract, however, the information of the molecules in the tentacle extract should be presented for the reproducibility. For example is it specific for only Cyanea capillata species? The present reviewer assume that all the jelly fish may has the similar capacity since they have toxic tentacle to date.Specific profile and molecular determination of the actin compounds will be appreciated.

Without the fractionation and isolation of the molecules, it is difficult to justify their findings for generalization.

---

## Round 0.3 · accepted · Accept

· Academic Editor

Accept

Congraturations!

Your revision is acceptable for publication in PeerJ. Readers will enjoy reading your new publication.

Thank you

Best regards
Editor

Reviewer 2 ·

Basic reporting

The change made in the discussion allows a better understanding of the problem.

Experimental design

The topic is of particular interest. The research group's commitment is intense, however, a few suggestion for a better understanding of the biological model were not followed.

Validity of the findings

no comment

Additional comments

no comment